# Genomic Evolution and Transcriptional Changes in the Evolution of Prostate Cancer into Neuroendocrine and Ductal Carcinoma Types

**DOI:** 10.3390/ijms241612722

**Published:** 2023-08-12

**Authors:** Srinivasa R. Rao, Andrew Protheroe, Lucia Cerundolo, David Maldonado-Perez, Lisa Browning, Alastair D. Lamb, Richard J. Bryant, Ian G. Mills, Dan J. Woodcock, Freddie C. Hamdy, Ian P. M. Tomlinson, Clare Verrill

**Affiliations:** 1Nuffield Department of Surgical Sciences, University of Oxford, Oxford OX3 9DU, UK; srinivasa.rao@nds.ox.ac.uk (S.R.R.);; 2Oxford Centre for Histopathology Research, University of Oxford, Oxford OX3 9DU, UK; 3Department of Oncology, University of Oxford, Oxford OX3 7DQ, UK; 4NIHR Oxford Biomedical Research Centre, University of Oxford, Oxford OX3 9DU, UK

**Keywords:** prostate cancer, neuroendocrine, small cell, whole genome duplication, tumor evolution, genomics

## Abstract

Prostate cancer is typically of acinar adenocarcinoma type but can occasionally present as neuroendocrine and/or ductal type carcinoma. These are associated with clinically aggressive disease, and the former often arises on a background of androgen deprivation therapy, although it can also arise de novo. Two prostate cancer cases were sequenced by exome capture from archival tissue. Case 1 was de novo small cell neuroendocrine carcinoma and ductal adenocarcinoma with three longitudinal samples over 5 years. Case 2 was a single time point after the development of treatment-related neuroendocrine prostate carcinoma. Case 1 showed whole genome doubling in all samples and focal amplification of AR in all samples except the first time point. Phylogenetic analysis revealed a common ancestry for ductal and small cell carcinoma. Case 2 showed 13q loss (involving RB1) in both adenocarcinoma and small cell carcinoma regions, and 3p gain, 4p loss, and 17p loss (involving TP53) in the latter. By using highly curated samples, we demonstrate for the first time that small-cell neuroendocrine and ductal prostatic carcinoma can have a common ancestry. We highlight whole genome doubling in a patient with prostate cancer relapse, reinforcing its poor prognostic nature.

## 1. Introduction

Prostate cancer is the most common cancer in men in the UK [1], and most are acinar adenocarcinoma type. It is an androgen-driven disease with signaling through the androgen receptor (AR), and although androgen deprivation therapy (ADT) can control tumor growth, many patients go on to develop castrate-resistant prostate cancer. The majority of prostatic neuroendocrine carcinomas develop in this setting and are termed treatment-related neuroendocrine prostatic carcinomas (t-NEPC) [2]. However, rarely de novo cases may also occur [3]. De novo small cell carcinoma (SCNEC) accounted for 0.03% of prostatic carcinomas in one analysis [4], and the less common large cell (large cell neuroendocrine carcinoma) may also occur. In the later stages of prostate cancer, t-NEPC occurs in up to 15–20% of patients treated with ADT [5]. The diagnosis is made on the basis of typical morphological features together with confirmatory immunohistochemical stains (e.g., Synaptophysin, Chromogranin) [6]. 

The origin of neuroendocrine prostatic carcinoma (NEPC), whether this is from the transdifferentiation of adenocarcinomas or oncogenic mutation of normal neuroendocrine cells, remains controversial, and the possible theories have been previously summarised by Shui et al. [6]. Previous studies have looked at the genomic and epigenetic dysregulation that takes place in this process, involving RB1, TP53, PTEN deletion, and MYCN and AURKA overexpression [7,8,9,10]. 

Prostatic ductal adenocarcinoma is a rare type of prostate cancer, but it is the second most common subtype of prostate cancer behind acinar adenocarcinoma, and on meta-analysis, 0.17% of cases were shown to be pure ductal type [11]. It is more commonly mixed with acinar adenocarcinoma, being present in 2.6% of prostatic adenocarcinomas [12] with a significantly higher risk of mortality than acinar adenocarcinoma [13]. It has a typical columnar cell type and often shows cribriform morphology. Opinion is divided as to the underlying biology (Ranasinha et al., as above).

In this study, we present 2 cases of NEPC that highlight important genomic changes in its evolution—the first case being de novo SCNEC and with sequential samples over 5 years, and for comparison, the second is a case of known advanced prostate cancer with the development of NEPC, presumed from the clinical information available to be t-NEPC. With these samples, we aimed to assess tumor evolution in archival prostate cancer samples to understand the distinct genomic changes present in morphologically different regions (small cell neuroendocrine cancer vs. ductal carcinoma, small cell neuroendocrine cancer vs. adenocarcinoma) within the same patients.

## 2. Results

### 2.1. A Case of de Novo SCNEC with Recurrence and Co-Occurrence of Ductal Adenocarcinoma

Case 1 consisted of four sequenced regions isolated from three sections from different time points (Figure 1A). SCNEC was identified at all three time points (NE_2012, NE_2015, NE_2017). A high-magnification view of NE_2012 is shown (Figure 1B). In the resected tissue from 2015, we observed prostate cancer with both SCNEC and ductal morphology (Figure 1C). Positive staining for Synaptophysin in the small cell region and the absence of staining in the ductal region served as confirmation of the phenotypes (Figure 1D). Higher power images of the ductal and SCNEC regions are also shown (Figure 1E,F). The ductal adenocarcinoma was composed mostly of papillary structures and complex glands with a lesser number of cribriform glands, all lined by tall columnar pseudostratified cells. Morphologically, the ductal adenocarcinoma was invasive rather than intraductal spread with some areas of comedo necrosis and Gleason patterns 4 and 5 [2]. 

### 2.2. Exome Sequencing Reveals Copy Number Changes in the Recurrent SCNEC

In order to determine genomic differences between different samples, we performed whole exome sequencing. Copy number analysis revealed complex profiles with evidence of whole genome duplication (WGD, defined as ploidy > 3) in all the samples (Figure 2B). In addition, a missense SNV in PTEN was also seen in all samples at a cancer cell fraction of ~1 (i.e., 100% of the cells have this specific point mutation in PTEN). These findings, along with several other shared SNVs between the different samples, pointed to a shared origin for all samples and confirmed the latter samples as recurrences from the original cancer. We identified a focal amplification of the androgen receptor gene in all except the earliest sample, timing this event between 2012 and 2015 (Figure 2C).

### 2.3. Phylogenetic Analysis Shows Evolution of the Cancer over Time

To further understand the sequence of molecular events between the original cancer and the recurrence in 2015 and 2017, we performed phylogenetic analysis using Dirichlet Process clustering of the copy number corrected SNVs. A number of shared SNVs (53, including PTEN, POLR3D, STAG2) with an average cancer cell fraction of ~1 suggest cluster A as the Most Recent Common Ancestor (MRCA) (Figure 3 and Appendix A). As the focal amplification at the AR gene locus is not present in the original cancer in 2012 but shared by all subsequent samples, it must have occurred in the intervening period between 2012 and 2015.

The ductal sample from 2015 (DUCTAL_2015) has several copy number changes unique to it that are not present in the SCNEC samples—e.g., 1p LOH, 6q amp. The SCNEC from 2017 (NE_2017) is distinguished by a cluster of 34 SNVs unique to it (including missense mutations in RYR2, LRP4, MPZL1, SLC1A1) (Figure 3).

### 2.4. Transcriptomic Analysis Reveals Differences in Gene Expression between Different Time Points and Morphologies

A 12-fold and 7-fold increase in AR expression was seen in NE_2015 and NE_2017, respectively, compared to NE_2012 (Figure 4A and Appendix A), matching the focal amplification seen in the copy number analysis. Gene ontology analysis (Figure 4B) shows the involvement of the Hallmark EMT pathway. Transcriptomic analysis revealed a higher AR, PSA, and SCHLAP1 gene expression in DUCTAL_2015 compared to NE_2015 (Figure 4C and Appendix A). There was no evidence of ERG fusion transcripts. No difference in FOLH1 (PSMA) expression was observed in NE_2017 vs. NE_2012 and NE_2015 vs. NE_2012 (|log2FoldChange| < 1).

### 2.5. Exome Analysis in a Second Patient Shows Evolution of t-NEPC

In case 2, containing Trans-Urethral Resection of Prostate chippings from a single time point, prostate cancer was seen with two distinct morphologies, acinar adenocarcinoma and t-NEPC (Figure 5A), following a prior diagnosis of acinar adenocarcinoma on prostate biopsy. Unfortunately, due to the time elapsed since diagnosis, more detailed information on the clinical or treatment history in this case could not be found. However, advanced prostate cancer is typically treated with hormonal therapy, and thus, the presumption was made that this was a t-NEPC, a diagnosis of de novo SCNEC being precluded by the prior diagnosis of acinar adenocarcinoma. This was further supported by both acinar adenocarcinoma and NEPC elements being present, which is a very typical pattern of development of t-NEPC. Copy number analysis identified RB1 loss (due to loss of heterozygosity of 13q) in both morphology regions (Figure 5B). A number of shared SNVs and CNAs confirm a common ancestry for both histological subtypes in this patient (Figure 5C and Appendix A). The presence of specific CNAs (3p gain, 4p LOH, 17p LOH involving TP53) in the t-NEPC region but not in the adenocarcinoma region suggests that the t-NEPC cells were derived from the adenocarcinoma.

## 3. Discussion

In this paper, we present novel insight into important genomic evolutionary changes over time in the development of NEPC and ductal adenocarcinoma in two patients by using highly curated samples. Case 1 showed genomic evolution over a period of 5 years in a de novo SCNEC. We identified the phenomenon of Whole genome Duplication (WGD) and, for the first time, a common ancestry for SCNEC and ductal adenocarcinoma. Case 2, by comparison, was t-NEPC and showed previously described changes in RB1 and TP53 loss. 

WGD in cancer is a mechanism by which cancer cells are thought to acquire resilience to deleterious mutations. Multiple copies of each gene result in a greater ability to withstand losses of essential genes [14]. It has been reported that less than 25% of prostate adenocarcinomas exhibit WGD in a pan-cancer analysis of whole genome (PCAWG) data [15], and this can be as high as 50% in metastatic samples [16]. In our previous work, we identified WGD as a potential driver of lymph node metastasis [17]. However, it is unclear how prevalent WGD is in histological subtypes of prostate cancer. Beltran et al. noted that there is no significant difference in the prevalence of polyploidy between castration-resistant neuroendocrine and adenocarcinoma of the prostate [8]. In this study, we show that WGD was present in the original cancer within SCNEC of the prostate. 

The focal amplification observed in the AR locus post-2012 may have been in response to hormone treatment. Beltran et al. reported that there is no significant difference between AR amplifications in castration-resistant neuroendocrine and adenocarcinomas of the prostate [8]. Hence, the AR amplification may simply be a common resistance mechanism to anti-androgen treatment. It is known that AR gene amplification is the most common reason for increased AR gene expression [18].

Transcriptomic analysis revealed increased AR mRNA expression relative to that in the 2012 sample, which is likely a result of AR amplification. AR expression is also higher in the ductal carcinoma region relative to the SCNEC region.

Immunohistochemistry for AR expression showed heterogeneous expression (Appendix A). While the intensity of staining is roughly similar to the pattern of AR expression observed in the RNA-seq data (NE_2012 < NE_2017 < NE_2015) in some regions, the staining is heterogeneous throughout the tissue, with distinctly low AR expression, particularly in some regions of the NE_2015 and DUCTAL_2015 samples. However, this heterogeneity within the NE_2015 and DUCTAL_2015 samples was not captured in the exome sequencing data, perhaps due to the low power to call subclones compared to whole genome sequencing. Novel spatial genomic techniques [19] could shine more light on complex samples such as these.

PSMA (FOLH1) expression is reported to be suppressed in prostate cancer following neuroendocrine differentiation [20]. However, we were unable to test this in our study due to the unavailability of an adenocarcinoma sample from patient 1.

PTEN loss is also associated with adverse prognosis and was identified in all samples from case 1, across ductal and small cell NE elements, but was not seen in case 2. Case 2, being t-NEPC, showed P53 and RB1 loss and the combination of RB1 loss and TP53 mutation or deletion occurs more commonly in NEPC (approximately 50%) than in prostatic adenocarcinoma (approximately 14%) [6,9,10]. We show here that in this case, RB1 loss is present in the MRCA, whereas TP53 loss was found exclusively in the small cell NEPC region. Analysis of the timing of RB1 and TP53 mutations in a larger number of cases may reveal the sequence of genetic changes necessary for this transformation.

This study highlights the insights that can be drawn from histology-guided multi-region genetic analysis in individual patients. We identified somatic mutations, transcriptomic changes, and tumor evolution in histologically distinct regions from limited archival samples collected across time points. Phylogenetic analysis enabled us to associate specific genetic changes with histological transformation and helped us to establish the sequence of these genetic changes. We have previously reported histology-guided genomic analysis in a case of amphicrine carcinoma (a rare type of prostatic adenocarcinoma showing both exocrine and NE differentiation) developing from acinar adenocarcinoma [17]. While data from only two patients cannot be generalized, the ability to identify mutations specific to different histological subtypes within the same patient can help with the selection of an appropriate combination of therapies to precisely target them.

## 4. Materials and Methods

This study was conducted under the Oxford Radcliffe Biobank research ethics approval (reference 19/SC/0173) with appropriate consent. The clinical details and tissue details are shown in Table 1. Case 1 died of prostate cancer (prostate cancer-specific mortality). The clinical status of case 2 is unknown but presumed to be prostate cancer-specific mortality.

### 4.1. Immunohistochemistry

Deparaffinized and rehydrated formalin-fixed paraffin-embedded sections were treated with 3% H_2_O_2_ to neutralize endogenous peroxidase. Antigen was retrieved using citrate buffer pH6. Background staining was blocked using 5% NGS/PBS. Primary antibody (Table 2) diluted with 5% NGS/PBS was applied to the samples overnight at 4 °C. Detection was performed with Biotinylated goat anti-mouse IgG (Vectorlabs BA-9200), ABC reagent (Vectorlabs PK-7100), and DAB Substrate Kit SK-4100 (Vectorlabs) sequentially. Counterstain was Harris’s hematoxylin.

### 4.2. Sample Processing

Distinct regions of cancer subtypes (acinar adenocarcinoma, neuroendocrine, ductal carcinoma) were annotated by a board-certified pathologist (CV). These regions were manually macrodissected and pooled from 5× 4um FFPE tissue sections. DNA was isolated from macrodissected tissue using the High Pure FFPET DNA and High Pure RNA Isolation Kits (Roche), as reported previously [21]. There was insufficient acinar adenocarcinoma in case 1 for macrodissection or sequencing. 

### 4.3. Exome Sequencing

Selection of exonic regions was performed from 100 ng of DNA using the TruSeq Exome Kit (Illumina), as reported previously [21]. The exome library was sequenced on the Illumina NextSeq 500 to a median depth of coverage of 89× (43×–217×) as 75bp paired-end reads. 

### 4.4. Pre-Processing

DNA sequences were trimmed using BBTools (v38.79-0) [22] and aligned to the human genome (hg38) using BWA-MEM (v0.7.17) [23]. Data were further pre-processed using the GATK Best Practices workflow to perform Base Quality Recalibration and PCR deduplication [24].

### 4.5. SNV Calling

Point mutations and short indels were called using Mutect2 (GATK v4.1.4.0) using multi-tumor mode and filtered for an F Score beta value of 0.005.

### 4.6. Copy Number Calling

ASCAT (v3) [25] was used to call the copy number profiles. 

### 4.7. Phylogenetic Tree Construction

Both CNAs and SNVs were taken into account to infer the phylogeny of the cancer in Case 1, with the former given precedence. SNVs were clustered using a Dirichlet process algorithm from Scikit-learn (v1.2.2). 

### 4.8. RNA Sequencing

RNA sequencing libraries were prepared using the NEBNext Ultra II RNA Library Prep Kit for Illumina (New England Biolabs, Ipswich, MA, USA), and three replicates from each sample were sequenced on the Illumina NextSeq 500 as 75bp paired-end reads. Reads were aligned to the reference human genome (hg38) using the STAR aligner [26], and differential expression analysis was performed using DESeq2(v1.38.3) [27]. Gene set enrichment analysis was performed with a pre-ranked gene list (signed log2FoldChange x adjusted *p* value) from the differential expression analysis using the fgsea R package(v1.24) [28].

## Figures and Tables

**Figure 1 ijms-24-12722-f001:**
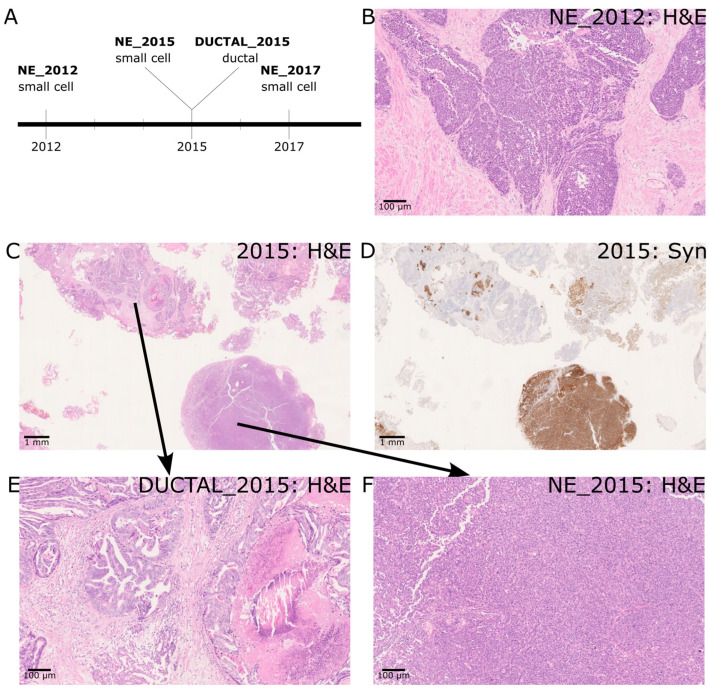
(**A**) Timeline of sample collection and histology for case 1. (**B**) 2012 histology—de novo small cell neuroendocrine prostatic carcinoma (9×) (**C**) 2015 histology with acinar, ductal and small cell neuroendocrine prostate carcinoma areas. (1×) (**D**) Small cell neuroendocrine prostate carcinoma areas with Synaptophysin positivity (1×). Ductal carcinoma areas are negative. (**E**) Higher power ductal carcinoma areas (9×) (**F**) Higher power small cell neuroendocrine areas (9×).

**Figure 2 ijms-24-12722-f002:**
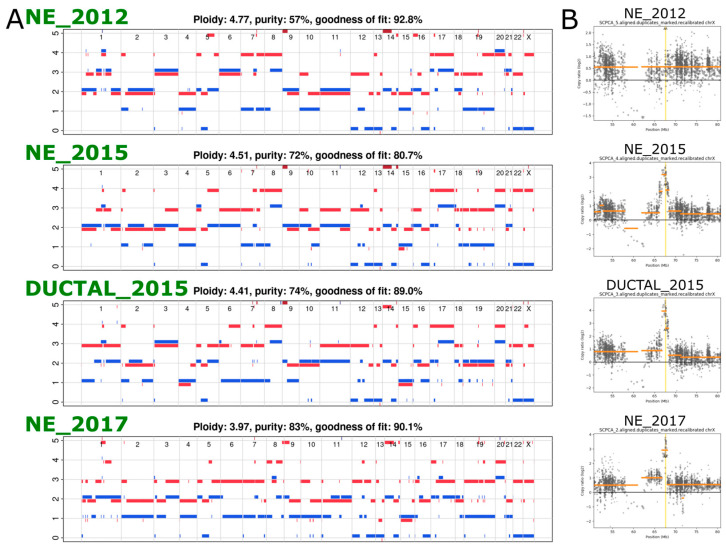
(**A**) Copy number profiles of the four samples from case 1. (**B**) Scatterplots of a region of chrX spanning the AR gene (locus indicated by the vertical yellow line), with the log2 read depth ratio (compared to the non-cancer sample) along the *y* axis.

**Figure 3 ijms-24-12722-f003:**
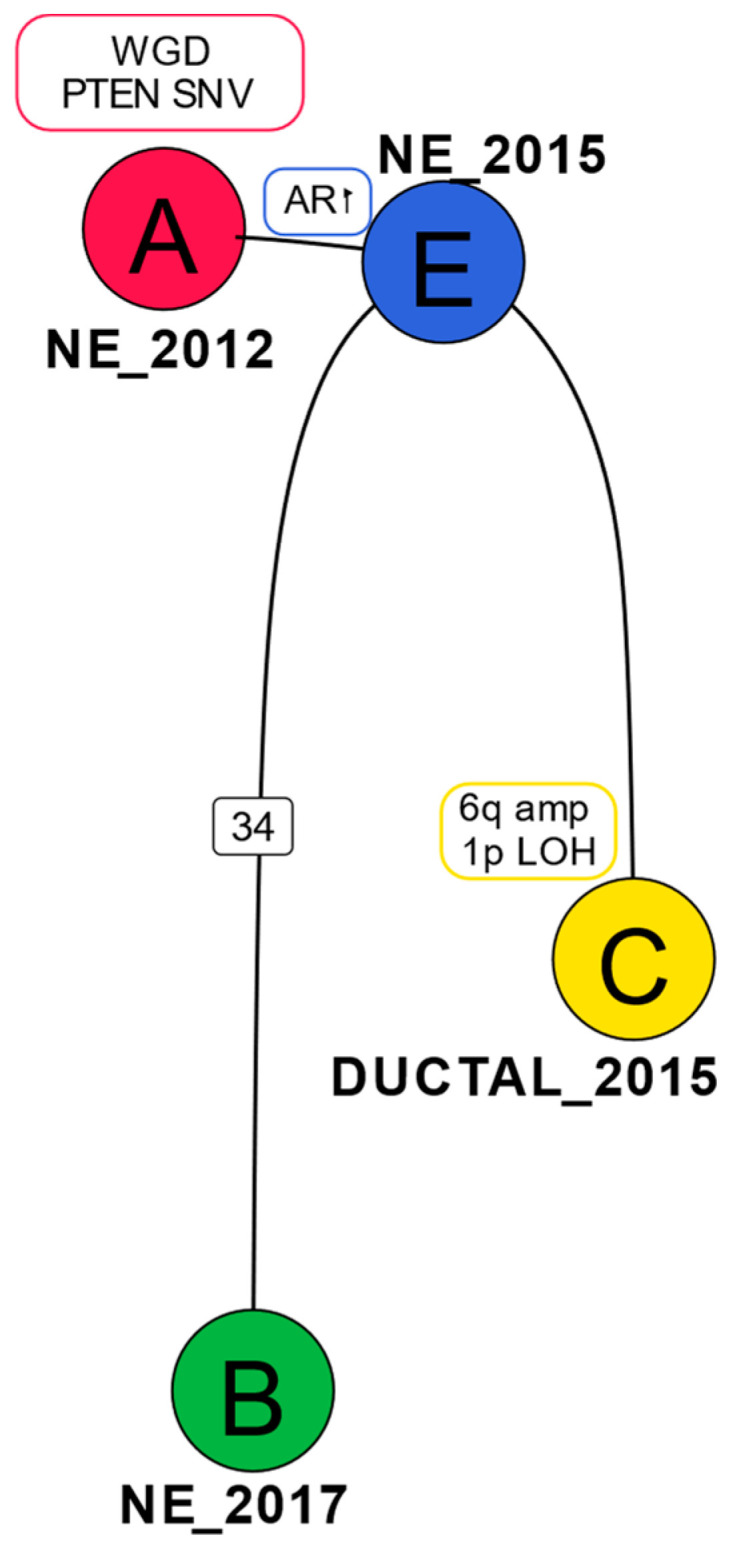
Phylogenetic tree for case 1: Genomic changes or number of SNVs (in the case of clone B) belonging to a specific clone are represented on the tree edges. The samples in which each of the clones were predominantly found are annotated close to the clones.

**Figure 4 ijms-24-12722-f004:**
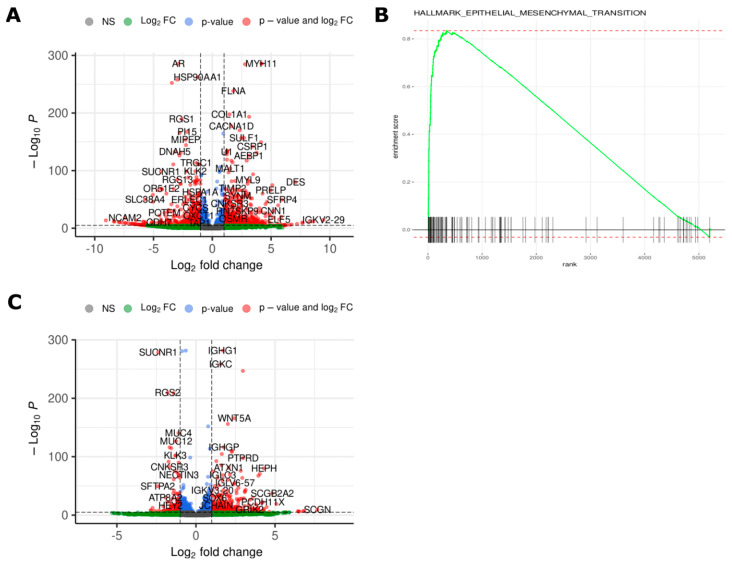
(**A**) Volcano plot showing genes differentially expressed between NE_2012 and NE_2017 (log2 fold change > 0 represents high expression in NE_2012). (**B**) Gene set enrichment analysis of differentially expressed genes between NE_2012 and NE_2017 revealed enrichment of the Epithelial-Mesenchymal Transition Hallmark Gene Set in NE_2017. (**C**) Volcano plot showing genes differentially expressed between DUCTAL_2015 and NE_2015 (log2 fold change > 0 represents high expression in DUCTAL_2015).

**Figure 5 ijms-24-12722-f005:**
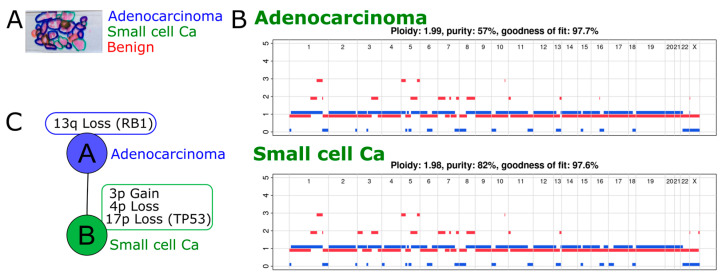
(**A**) Prostate cancer chippings from one-time point from case 2 showing regions marked for adenocarcinoma and treatment-related neuroendocrine prostate carcinoma. (**B**) Copy number profiles for the two morphologically distinct regions. (**C**) Phylogenetic relationship between the adenocarcinoma and treatment-related neuroendocrine prostate carcinoma regions.

**Table 1 ijms-24-12722-t001:** Clinical and specimen details of the two sequenced prostate cancer cases. Case 1 features longitudinal samples of tissue over a 5-year period consisting of small-cell neuroendocrine carcinoma and ductal-type adenocarcinoma. Case 2 samples are from a single time point consisting of acinar adenocarcinoma and treatment-related neuroendocrine prostatic carcinoma. The histological terminology used was taken from guidance in place at the time (all samples prior to WHO 2022). Note that essential criteria for small cell neuroendocrine carcinoma are morphological, with positive immunostaining for neuroendocrine markers being desirable [2].

Histology	Clinical Details Provided with Histology Section	IHC Profile/Morphology	Diagnosis
Case 1			
Section 1 (2012)	Prostate chippings. Recently started hormone therapy for node-positive cancer, thought to be prostate.	PSA+ TTF-1+ Synaptophysin– CD56– ChromograninA– Strong morphological impression of SCNEC	Small cell carcinoma of prostatic origin.
Section 2 (2015)	Prostate chippings. Prostate cancer on hormones and chemotherapy. Post radiotherapy.	In NE areas: PSA not performed TTF-1+ Synatophysin+ CD56+ ChromograninA+	Prostatic adenocarcinoma of acinar and ductal type with areas of small cell (neuroendocrine differentiation)
Section 3 (2017)	History of prostate cancer with neuroendocrine component—due to start chemotherapy. Visible haematuria, new bladder mass on flexible cystoscopy. TURBT—mass connected to the prostate—likely extension of prostate cancer rather than new bladder tumor. On Zoladex.	PSA + (patchy) TTF-1- Synaptophysin—(occasional cells + only) CD56– ChromograninA– Strong morphological impression of SCNEC.	Bladder and prostate chippings—high-grade prostatic adenocarcinoma with small cell carcinoma/small cell neuroendocrine differentiation
Case 2			
1999	Known advanced prostate cancer (diagnosed 1998 on prostate biopsy as Gleason Score 8 (4 + 4) adenocarcinoma.	Strong morphological impression of SCNEC	Prostate adenocarcinoma with many small cell areas

**Table 2 ijms-24-12722-t002:** Antibodies used for immunohistochemistry.

Protein	Antibody Type	Supplier and Usage Details
Chromogranin A	mouse monoclonal antibody clone DAK-A3	DAKO M0869 (working dilution 1/100)
Synaptophysin	mouse monoclonal antibody	Novocastra NCL-L-SYNAP-299 (working dilution 1/100)
CD56	mouse monoclonal antibody	Novocastra NCL-L-CD56-504 (working dilution 1/100)
Androgen Receptor	mouse monoclonal antibody	Leica AR-318-L-CE (working dilution 1/25)

## Data Availability

All raw sequencing data was submitted to the European Genome-Phenome Archive (EGA) under the dataset IDs EGAD00001011149 (exome) and EGAD00001011154 (RNASeq).

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
