# Peer review of "Genomic Evolution and Transcriptional Changes in the Evolution of Prostate Cancer into Neuroendocrine and Ductal Carcinoma Types"

_ijms, 2023, doi:10.3390/ijms241612722_

Round 1

Reviewer 1 Report (Previous Reviewer 2)

In this revision, the authors have provided more information on the cases studied by adding detailed explanations and executing AR IHC. Therefore, I can recommend this work to be considered for publication in IJMS under the category of Case Report articles, subject to the following minor revisions:

1- In the caption of Fig. S9, please comment on the ratio of nuclear and cytoplasmic AR positivity in each sample, as the magnification is not high enough to evaluate the localization of AR staining.

2- From the angle of PSMA-targeted therapy, please comment on the expression value of FOLH1, especially as it has not changed in NE_2017 vs NE_2012 and NE_2015 vs NE_2012. Is the expression high or low? Then, add a few lines of discussion considering the available literature on PSMA positivity or suppression in NEPC.

3- This reviewer was unable to access the shared submitted data on the European Genome-Phenome Archive (EGA) as EGAD00001011149 (exome) and EGAD00001011154 (RNASeq). Please ensure that the files are available.

Author Response

1- In the caption of Fig. S9, please comment on the ratio of nuclear and cytoplasmic AR positivity in each sample, as the magnification is not high enough to evaluate the localization of AR staining.

We presume the reviewer meant Fig S6 (the last figure in the supplementary file). We added this text to the caption: “While the AR expression is nuclear in NE_2012, in the rest of the samples (NE_2015, DUCTAL_2015, NE_2017) there is cytoplasmic staining in all cells where there is nuclear staining.”

2- From the angle of PSMA-targeted therapy, please comment on the expression value of FOLH1, especially as it has not changed in NE_2017 vs NE_2012 and NE_2015 vs NE_2012. Is the expression high or low? Then, add a few lines of discussion considering the available literature on PSMA positivity or suppression in NEPC.

While it is difficult to comment on the absolute expression level, FOLH1 expression is high relative to GAPDH expression (mean normalised counts in all the samples: 1597.45 vs 769.86) and low relative to ACTB (1597.45 vs 2617.42). We cannot comment on PSMA suppression following neuroendocrine differentiation as we did not have an adenocarcinoma (pre-neuroendocrine) sample to compare with the NE samples.

We have added the following text in the discussion, lines 190-192:

“PSMA (FOLH1) expression is reported to be suppressed in prostate cancer following neuroendocrine differentiation[20]. However, we were unable to test this in our study due to the unavailability of an adenocarcinoma sample from patient 1.”

3- This reviewer was unable to access the shared submitted data on the European Genome-Phenome Archive (EGA) as EGAD00001011149 (exome) and EGAD00001011154 (RNASeq). Please ensure that the files are available.

We have now requested the EGA helpdesk to release the datasets, which should be actioned in the next few days.

Additional changes:

  • We inadvertently used the word ‘samples’ instead of ‘patients’ in our previous revision – this has now been corrected in line 209.
  • The differential expression analysis results and normalised counts data from RNASeq are now added as a supplementary file.

Reviewer 2 Report (Previous Reviewer 3)

I previously reviewed this article and have been asked to re-review the revision - I looked at the author responses and the revised manuscript and now believe that it warrants publication in IJMS. Thank you.

Author Response

Many thanks for your constructive comments which helped to improve the article. We made the following changes:

Additional changes:

  • We inadvertently used the word ‘samples’ instead of ‘patients’ in our previous revision – this has now been corrected in line 209.
  • The differential expression analysis results and normalised counts data from RNASeq are now added as a supplementary file.

This manuscript is a resubmission of an earlier submission. The following is a list of the peer review reports and author responses from that submission.

Round 1

Reviewer 1 Report

It is an interesting topic and the results can add valuable information to the cancer research field. 

Reviewer 2 Report

This manuscript presents two cases of advanced prostate cancers. The first case is described as a de novo NEPC, which demonstrates divergent clonal evolution to ductal carcinoma and AR-positive NEPC. RNA-seq was used to profile this case further, revealing the rescue of AR expression at the RNA level in NEPC. The second case is a mix of adenocarcinoma and NEPC. The authors speculate that the coexistence of Adeno and small cell in this case could be due to the treatment history, leading them to classify it as t-NEPC.

This reviewer found the report intriguing, and I believe it could be considered for publication following these revisions:

·      Page 5, Line 108: The authors reported a >4-fold increase in AR expression in NE_2017, as illustrated in the volcano plot in Fig. 4A. However, it might be beneficial to clarify that if the log2-fold change in expression is 4, the actual change could exceed 16-fold.

·      Regarding the above statement, do you consider NE_2017 as an AR-positive advanced cancer exhibiting NE-positivity? Is there any possibility that NE_2012 was also a mixed case with heterogeneity, and NE_2017 is a minor subset of NE_2012? Or, is there any correlation between the treatment and rescue of AR expression?

·      Do you have any AR IHC on these presumably AR-positive tissues to discuss AR-localization and potentially support the lack of AR-signaling?

·      In light of new knowledge about possible PSMA expression in AR-positive and AR-negative NEPC (Beltran 2023, Nature Cancer), please comment on the FOLH1 expression in samples with RNA data. 

·      Please provide more detailed information about the treatment history of case 2.

·      It would be beneficial to submit the sequencing data to the EGA prior to publication.

·      For the Supplemental Data Heatmaps: Please add information about the color scale bars. Do they represent normalized log2 expression (TPM/...)?

·      Line 38: t-rNEPC should read t-NEPC

Reviewer 3 Report

This is a well written manuscript, and the subject area is of importance. A major concern is the inclusion of only 2 patient sample. While the samples are highly curated and the data are of interest, in my opinion it is not appropriate to make inferences based on these data – as is, the manuscript is more comparable to a clinical case report and the data are descriptive only. How this study differs from similar (and larger) studies is also unclear. My other concerns and some comments are listed below:

1.       Abstract, lines 26-27: I think this sentence needs to be rephrased to align with the fact that only 2 cases are being assessed/it’s not appropriate to make inferences. Perhaps something along the lines of ‘our data indicate that it is possible small cell neuroendocrine and ductal prostatic carcinoma have a common ancestry’.

2.       Introduction, line 37: Please include a citation for this statement.

3.       Introduction, lines 37-41: Please state % patients who develop neuroendocrine cancers following ADT as well as the % patients who present with de novo neuroendocrine cancers and provide citations.

4.       Introduction, lines 44-49: Please provide a rationale for conducting the current study in this paragraph – what does this study do that the other studies that you cite have not done?

5.       Introduction, lines 50-52: Please state % patients who develop ductal adenocarcinoma and provide a citation. Does this typically develop de novo or in response to ADT?

6.       Discussion: More detailed comparisons need to be made with existing/similar studies which have been conducted. For example those mentioned in the Introduction (RB1, TP53, PTEN deletion and MYCN and 48 AURKA [5–8]). Do the data align?